# Fall Risk Assessment and Prevention Strategies in Nursing Homes: A Narrative Review

**DOI:** 10.3390/healthcare13040357

**Published:** 2025-02-07

**Authors:** Takeshi Miura, Yuka Kanoya

**Affiliations:** Department of Gerontological Nursing, Nursing Course, School of Medicine, Yokohama City University, Yokohama 236-0004, Japan; ykano@yokohama-cu.ac.jp

**Keywords:** nursing homes, long-term care, falls, risk assessment, fall prevention

## Abstract

**Abstract: Background/Objectives:** Falls in nursing homes significantly affect residents’ health and quality of life. Although considerable progress has been made in fall prevention strategies in acute care settings and community environments, research on fall risk assessment methods and prevention strategies in nursing homes remains scarce. Nursing homes provide long-term care for residents with high levels of dependency, presenting unique challenges in managing fall risks. Nevertheless, unlike hospitals, nursing homes face operational constraints, such as limited resources and staffing. These factors necessitate a tailored approach to fall risk management. This study aimed to summarize the current knowledge of fall risk assessment and prevention methods in nursing homes, clarify practical insights for implementation, and identify research gaps based on studies published over the past five years. **Methods:** This narrative review targeted studies published between 2019 and 2024 on fall risk assessment and prevention methods in nursing homes. A literature search was conducted using the PubMed and CINAHL databases, combining keywords such as “Accident Prevention”, “Fall Risk Assessment”, “Nursing Homes”, “Long-Term Care”, and “Aged”. The inclusion criteria allowed the inclusion of peer-reviewed academic articles on fall risk assessment or prevention interventions in long-term care facilities published in English within the past five years. Studies focusing on community-dwelling older adults, hospitalized older adults, and review articles were excluded. **Results:** This review analyzed 55 studies; among them, 27 studies focused on fall risk assessment and 28 focused on fall prevention. Regarding fall risk assessment, widely used tools, such as the Morse Fall Scale, which is also utilized in medical settings, have been extensively examined. In addition, new predictive methods utilizing electronic health records (EHR) and wearable devices have been introduced. However, the limited number of reports highlights the potential challenges in developing indicators that consider the unique characteristics and feasibility of LTC facilities. Regarding fall prevention, studies have examined indirect approaches, such as environmental modifications, and direct interventions, such as exercise programs. Furthermore, staff education and organizational initiatives are crucial in implementing preventive measures. However, most studies have been conducted in experimental settings, with limited empirical research available to assess the practical applications of these strategies in real-world nursing home environments. **Conclusions:** Fall risk assessments in nursing homes lack practical indicators tailored to the specific characteristics of long-term care facilities. Although various digital technologies have been explored for fall prevention, empirical studies that validate their real-world applicability are lacking.

## 1. Introduction

The global population of individuals aged 65 years and older is rapidly increasing compared to other age groups, primarily due to rising life expectancies [1,2,3]. By 2050, the population of people aged 65 and older is expected to double, while the population of those aged 80 is projected to triple [1]. Older adults, commonly defined as those aged 65 years and older, are at a higher risk of falls and hospitalization compared to other age groups, and the number of older individuals residing in nursing homes is expected to increase in the coming years [4].

Falls are a significant concern in nursing homes, where residents often experience frailty, limited mobility, and multiple chronic conditions [5]. Falls are one of the leading causes of injury and mortality among older adults globally, particularly in institutional care settings such as nursing homes [6]. Research has shown that falls in nursing homes can result in severe physical injuries, a decline in quality of life, and increased healthcare costs [5,6,7,8]. Despite developing various fall prevention strategies in hospital and community settings [9,10,11,12], there remains a lack of clear and consistent evidence regarding fall prevention in long-term care facilities, particularly those serving highly dependent older adults [13].

The rising incidence of falls in nursing homes imposes a significant burden on residents, healthcare staff, and the overall healthcare system [14]. Nursing homes are designed to provide care for older adults with high dependency levels, presenting unique challenges in managing fall risks [15]. The incidence of falls in long-term care facilities is significantly higher than that in community-dwelling older adults, with an average of 1.7 falls per resident annually, compared to 0.65 falls in private homes. Furthermore, 57.9% of nursing home residents experience at least one fall annually, and 24.2% of these cases result in hospitalization [13,16]. Unlike acute care hospitals, which provide short-term care, nursing homes offer long-term care, making fall prevention an ongoing concern. Practical fall risk assessment in such environments requires a comprehensive understanding of residents’ health conditions and the environmental factors contributing to falls. Additionally, fall prevention measures should be feasible and adaptable to the daily operations of nursing homes, where resources and staffing levels may limit the scope of interventions.

This narrative review aims to organize and summarize the current knowledge of fall risk assessment and prevention methods in nursing homes, clarify practical insights for implementation, and identify research gaps based on studies published over the past five years (2019–2024). The findings of this review will serve as a foundation for healthcare professionals, such as physicians and nurses working in elderly care facilities, enabling them to better understand their patients’ characteristics and challenges. Furthermore, this review highlights key areas requiring further research to enhance fall prevention strategies in nursing homes.

## 2. Methods

The literature search period for this review was set from 2019 to 2024, to incorporate the latest research findings in this field. The studies included in this review were selected based on their focus on fall risk assessment methods or fall prevention interventions in nursing homes. A comprehensive literature search was conducted using Medline via PubMed and the CINAHL database to identify relevant studies. The keywords used included combinations of terms such as Accident Prevention, Fall Risk Assessment, Nursing Homes, Long-Term Care, and Aged (Table 1). The researchers developed a draft search strategy for Medline via PubMed in consultation with a medical information specialist. Subsequently, a search strategy for the CINAHL database was constructed based on the search queries used in PubMed (Table 1).

The inclusion criteria for the studies were as follows: (1) peer-reviewed academic articles; (2) published within the past five years; (3) written in English; (4) focusing on long-term care facility settings; and (5) employing observational, interventional, or qualitative research designs. The exclusion criteria were as follows: (1) studies focusing solely on community-dwelling older adults; (2) studies focusing on hospitalized older adults; and (3) systematic or scoping reviews.

A literature search was conducted using the search strategies outlined in Table 1. The initial search yielded 598 articles. The titles and abstracts of these articles were screened to determine their relevance to the scope of this review. As a result, 141, 100, and 55 articles were selected after title, abstract, and full-text review, respectively (Figure 1).

The eligible studies included in this review provided detailed descriptions of fall risk assessment tools and prevention strategies specific to nursing homes. The included studies were categorized based on the fall risk assessment and prevention methods on which they focused, organizing similar approaches. By examining these studies, this review clarifies the current trends in fall risk assessment and prevention in nursing homes and highlights their applicability, strengths, and challenges.

## 3. Results

The 55 reviewed studies were categorized into two main areas: fall risk assessment and fall prevention. Regarding risk assessment, 13 studies focused on risk assessment and indicators [17,18,19,20,21,22,23,24,25,26,27,28,29], three on data utilization [30,31,32], and 11 on the application of digital technologies [33,34,35,36,37,38,39,40,41,42,43]. Regarding fall prevention, six studies addressed environmental adjustments and safety measures [44,45,46,47,48,49], 17 focused on prevention programs [50,51,52,53,54,55,56,57,58,59,60,61,62,63,64,65,66], and five examined organizational management and staff education [67,68,69,70,71]. The Results section describes the characteristics, indicators, programs, and technologies discussed in these studies.

### 3.1. Fall Risk Assessment

#### 3.1.1. Risk Assessment and Indicators

The reviewed studies presented various approaches to assessing the fall risk among nursing home residents, ranging from traditional risk assessment tools to advanced assessment methods. Owing to their simplicity, traditional tools continue to be widely used. Among the tools reported were the Morse Fall Scale [18,29], Peninsula Health Falls Risk Assessment Tool [20], Toulouse Saint Louis University Mini Falls Assessment [21], InterRAI Fall Risk Clinical Assessment Protocol [24], Scott Fall Risk Screen [24], Modified Fall Risk Tool [24], Berg Balance Scale [25], Balance Evaluation Systems Test [25], Mini-BESTest [25], Brief-BESTest [25], STOP-FALLING checklist [26], and Injury Risk in Nursing Homes tool [28]. These tools assess mobility restriction, cognitive impairment, and medication use.

Additionally, studies have explored methods to assess fall risks based on the gait conditions of residents [17,27], using electronic health records to identify at-risk individuals [19]. Research evaluating gait patterns has emphasized their value in assessing fall risks, whereas electronic health record (EHR)-based methods have highlighted the potential to monitor and assess the risk dynamically by integrating data.

These findings highlight the importance of various approaches, ranging from foundational tools to cutting-edge technologies. Although traditional tools remain indispensable for baseline assessments, technologies such as gait analysis and assessment methods using EHRs are being explored to address the complex and ever-changing health conditions of nursing home residents. These findings can contribute to fall risk assessments tailored to the characteristics of nursing home residents and to the selection of effective intervention strategies based on these assessments.

#### 3.1.2. Data Utilization

The use of EHRs and large-scale data for fall risk assessment has garnered increased attention. Routinely collected EHRs can serve as dynamic and comprehensive tools for risk assessment and management in nursing homes. The reviewed studies have investigated fall risk assessment models using routinely collected data and administrative datasets [30,31,32].

A study using EHR data developed and internally validated a dynamic fall risk assessment and monitoring tool in aged care settings, focusing on methods of dynamically evaluating the constructed models [30]. Another study utilized the Minimum Data Set (MDS) from nursing homes to develop a fall risk assessment model. This model incorporated factors such as medication data, the use of walking aids, and physical function to assess the fall risk within 90 days [31]. Furthermore, another study leveraging healthcare system data measured fall-related injuries and developed measurement models [32].

Despite these advancements, some challenges must still be overcome when implementing EHR-based risk assessment models. Many nursing homes lack the infrastructure to utilize these systems fully, and staff training is critical to ensure accurate data entry and interpretation. Moreover, addressing privacy and data security concerns is essential to building trust and ensuring the ethical use of resident information. The reviewed studies emphasized the value of routinely collected EHR data in developing dynamic, accurate, and scalable fall risk assessment tools. Nursing homes can effectively enhance their capacity to assess and manage fall risks by utilizing existing datasets and advanced measurement models. However, the limited number of studies and the remaining challenges indicate that further efforts are needed to develop practical methods that nursing professionals can readily adopt.

#### 3.1.3. Use of Digital Devices

The reviewed studies highlighted various digital devices used in fall risk assessment and prevention in nursing homes, demonstrating their potential to complement traditional methods through real-time monitoring and analytical capabilities, ultimately leading to more effective assessments by nursing staff. Wearable devices have been a common focus, with reports on their ability to evaluate mobility and balance. For instance, a spatiotemporal gait analysis validated the G-STRIDE inertial sensor for the assessment of fall risks [36]. Similarly, the Smart Belt demonstrated the potential to mitigate the impact of falls by monitoring movement patterns and deploying airbags to reduce hip injuries during falls [37]. Smartphone-based gait analysis systems such as the mVEGAS have also been validated in identifying mobility impairments in various settings [39]. Additionally, the MoTFall project showcased the integration of applications and sensors to address fall risks [40]. In Bangkok, a robotics-based fall prevention program improved knowledge, promoted exercise, and enhanced balance among older adults, thereby reducing fall incidents [43]. Furthermore, cost-effective activity trackers helped to track daily activities and fall risk indicators in resource-limited environments [41].

These studies illustrate the expanding role of wearable devices, sensors, and robotics in fall risk assessment and prevention in nursing homes. These devices can facilitate efficient nursing assessments, even with constrained resources in resident-centered care settings, where limited staff must manage many residents.

### 3.2. Fall Prevention Methods

#### 3.2.1. Environmental Adjustments and Safety

The reviewed studies highlighted a range of environmental adjustments and safety measures to reduce fall risks in nursing homes, emphasizing their impacts and practical implementation challenges. A study in Shanghai highlighted environmental hazards such as poor lighting and uneven floors as significant contributors to falls, advocating for systematic facility-wide improvements [47]. Upgraded lighting, including motion-sensitive systems, demonstrated significant fall rate reductions and boosted residents’ confidence in nighttime navigation [45]. Wearable safety devices such as hip airbags effectively reduce the severity of fall-related injuries in long-term care facilities [46,49]. However, varying perceptions regarding the use of hip protectors have also been revealed [48]. Specialized equipment such as lifting cushions has proven valuable in reducing complications and caregiver strain by assisting fallen residents [44]. These studies underscore the importance of environmental adjustments and targeted safety measures as foundational strategies for fall prevention in nursing homes.

#### 3.2.2. Fall Prevention Programs

In addition to programs with proven effectiveness, such as the Otago Exercise Program [65], new interventional programs, such as the Staying UpRight [58] and SUNBEAM programs [61], have shown the potential to improve residents’ stability. Tai Chi has also been suggested to reduce the fear of movement [55]. Furthermore, the effectiveness of volunteer-supported walking programs [57], leisure activities [62], and music-based programs [66] has been validated. Multifactorial interventions that combine various approaches have recently gained attention [54,64]. These exercise-integrated programs highlight the importance of preventive measures in communities, healthcare settings, and older adult care facilities.

#### 3.2.3. Organizational Management and Staff Education

The reviewed studies emphasized the importance of organizational management and staff education in fall prevention strategies within long-term care facilities, highlighting the roles of staff training, interdisciplinary collaboration, and leadership support. Frequent and concise training sessions (“brief but often”) were found effective in enhancing staff skills and maintaining engagement, particularly in the use of risk assessment tools and safe transfer techniques [67]. A cross-sectional study highlighted the variability in staff confidence levels, showing the need for targeted training to address gaps in the understanding of environmental hazards and monitoring technologies [68]. These measures have been linked to a reduction in fall incidents.

Interdisciplinary approaches such as the Training In Policy Studies program have demonstrated the benefits of personalized fall prevention strategies developed collaboratively by nurses, physiotherapists, and other professionals [69]. Integrating nurse practitioners into care facilities enhanced fall risk management, offering cost savings and improved resident outcomes through medication reviews and staff mentorship [70]. Ethical and operational challenges, such as staff acceptance of new technologies and privacy concerns, were also identified. Leadership commitment was critical in ensuring the smooth and ethical implementation of safety practices [71].

Overall, these studies highlight that effective fall prevention requires strong leadership, continuous education, and teamwork, as well as addressing operational and ethical challenges.

## 4. Discussion

### 4.1. Fall Risk Assessment

Fall risk assessment in nursing homes remains a critical priority because of its significant implications for residents’ health and quality of life. This review points out that research on fall risk assessment tools in nursing homes is limited compared to studies conducted in other settings, highlighting the need to address this gap. Traditional tools such as the Morse Fall Scale (MFS) and Peninsular Health Falls Risk Assessment Tool (PH-FRAT) continue to be widely utilized because of their simplicity and cost-effectiveness [18,20]. These tools evaluate essential factors, including mobility limitations, cognitive impairments, and medication use, providing a foundational framework for the identification of fall risks. However, their static nature and inability to capture daily fluctuations in the health conditions of residents, such as acute illnesses, medication side effects, and functional changes, pose significant limitations [72]. The reviewed studies also suggested that these tools may underperform in populations with specific needs, such as residents with cognitive impairments, indicating the need for tailored refinements [73]. Various fall risk assessment tools have been developed and validated for hospitalized patients and community-dwelling older adults; however, a review of studies from the past five years focusing on nursing homes revealed that reports specific to these settings still need to be made available. Further research is anticipated to support the development of more effective assessment tools tailored to the unique needs of nursing home residents [9,74,75,76].

In contrast to the extensive research in acute care hospitals, which often utilize large-scale healthcare databases to develop advanced assessment indicators, progress in nursing homes remains comparatively lacking [77,78,79]. This disparity is attributed to the need for standardized observational indicators specific to nursing homes and the absence of comprehensive databases. Bridging this gap requires the development of unified data-collection frameworks and investment in large-scale datasets tailored to nursing home populations. However, the implementation of such systems is hindered by challenges such as insufficient technological infrastructure in nursing homes, staff training needs, and concerns about data privacy and security [80].

Wearable devices and environmental sensors provide additional opportunities to enhance fall risk assessments. Devices such as the G-STRIDE inertial sensor, the Smart Belt, and smartphone-based systems enable the continuous monitoring of gait patterns, mobility, and physiological parameters [36,37]. These technologies complement EHR systems by adding real-time data to improve the assessment accuracy. However, their widespread adoption faces financial constraints, resident resistance, and maintenance challenges.

#### 4.1.1. Fall Prevention Methods

Fall prevention in nursing homes requires flexible strategies that integrate environmental, physical, and organizational approaches to address the diverse cognitive functions and daily life activities of residents.

Environmental modifications form the foundation of fall prevention [81]. Simple interventions can reduce fall rates by eliminating tripping hazards, installing handrails, and introducing motion-sensitive lighting [82]. Advanced measures, including non-slip flooring, automated bed sensors, and wearable devices such as hip airbags, can further mitigate the severity of fall-related injuries. Exercise programs also play a pivotal role in fall prevention [83,84]. Interventions such as the Otago Exercise Program [65] and SUNBEAM [61] have significantly improved the physical stability of residents and reduced the fall rates. Additionally, Tai Chi, which has been validated in numerous studies, effectively enhances balance, reduces fear of movement, and improves cognitive function [55]. This dual benefit is particularly advantageous for residents with mild cognitive impairment as it supports physical and mental health. Combining these programs with complementary strategies such as medication reviews and staff education further amplifies their effectiveness. However, individualized program designs are indispensable in addressing the diversity of physical and cognitive conditions among residents. Organizational strategies and staff education are critical in effectively implementing fall prevention measures [67,68,69,70,71,85]. Frequent and concise training sessions help to maintain staff knowledge and skills [67]. The implementation of such programs requires substantial resources. Collaboration with external organizations and community health initiatives can play a crucial role in overcoming these constraints. For example, involving physiotherapy students or volunteers can help to alleviate staff’s workloads while maintaining program quality [57]. In conclusion, fall prevention in nursing homes requires a comprehensive approach that integrates environmental adjustments, exercise programs, cognitive interventions, and organizational strategies. Although challenges related to resource limitations and resident acceptance persist, innovative and evidence-based solutions hold great potential in improving residents’ safety and quality of life. Future research should focus on developing scalable, sustainable, and individualized interventions to address these barriers effectively.

#### 4.1.2. Future Directions in Research

Research on fall risk assessment and prevention in care facilities faces numerous challenges but holds significant potential to improve safety and care quality. Several key directions for future research have been identified. First, it is essential to standardize fall risk assessment. The development of simple and universally applicable tools that can be seamlessly integrated into daily workflows is expected to ensure consistency in evaluations across facilities, enhance the assessment accuracy, and reduce staff workloads [86]. Data integration and infrastructure development are critical issues. Establishing centralized systems to aggregate and share data from care facilities would enable the efficient tracking of fall incidents and the practical analysis of interventional outcomes. Achieving this requires investments in data sharing infrastructure and the establishment of ethical governance frameworks. Furthermore, translating research findings into practical applications is vital. Although the effectiveness of many fall prevention strategies, such as exercise programs, medication reviews, and environmental modifications, has been demonstrated in controlled settings, adapting these strategies to real-world constraints necessitates the development of sustainable models. Moreover, a comprehensive approach that addresses the complex needs of the residents is crucial. The integration of the physical, cognitive, emotional, and social dimensions into prevention strategies can provide tailored care. Interdisciplinary collaboration among healthcare professionals is the key to delivering holistic care that meets the individual needs of residents [87]. Finally, technological innovation plays an important role. Wearable devices, EHRs, and environmental sensors offer opportunities for continuous monitoring and proactive interventions. However, challenges, such as costs, user resistance, and inadequate digital infrastructure, must be addressed. Collaboration between technology developers and care providers is essential in establishing practical and actionable interventional methods suitable for real-world implementation. Addressing these challenges can enhance the safety and efficacy of fall prevention in care facilities while aligning with advancements in other healthcare fields. Future research should prioritize the development of practical evidence-based assessment and prevention strategies that align with these directions to deliver high-quality care in care settings.

## 5. Conclusions

This review highlights the current state and challenges in fall risk assessment and prevention in nursing homes. Research on fall risk assessment remains insufficient in terms of considering the specific setting of this field while ensuring its feasibility. Moreover, the absence of standardized indicators and large-scale databases for verification poses a significant barrier to progress in this area. Future research should prioritize the development of fall risk assessment indicators tailored to the unique characteristics of long-term care facilities. These indicators should be designed to be practical and easy to understand, not only for nursing professionals but also for caregivers and a diverse range of healthcare and long-term care staff.

Existing evidence demonstrates the effectiveness of exercise interventions. However, considering the ongoing aging of the population and the increasing shortage of healthcare and caregiving staff, conventional approaches alone may not be sufficient as sustainable fall prevention strategies. Therefore, establishing more effective and efficient methods that incorporate digital technology is essential. Real-time monitoring using wearable devices and environmental sensors has the potential to facilitate the early detection of fall risks and enable individualized interventions, contributing to the optimal use of limited resources.

On the other hand, although the implementation of digital technologies is progressing to some extent, most existing studies have been conducted in experimental settings, raising concerns about the safety and effectiveness of these technologies in real-world applications. Future research should focus on promoting practical studies that evaluate these technologies in actual nursing home environments and incorporate the perspectives of both caregivers and residents. Furthermore, successful implementation requires improvements in technological infrastructure, enhanced staff training programs, and the establishment of frameworks that address data privacy and ethical considerations.

The findings of this review suggest that the key future directions for fall risk assessment and prevention in nursing homes include the development of evidence-based assessment tools, establishment of innovative approaches leveraging digital technologies, and promotion of practical studies to ensure their applicability in real-world settings. These efforts are expected to enhance the effectiveness of fall prevention measures and improve the safety and quality of life of nursing home residents.

### Limitation

This study had several limitations. Since the primary aim of this review was to present recent evidence to readers, this restricted the scope of the literature databases and the time frame considered. Consequently, relevant studies might have been overlooked. Additionally, it should be noted that evidence has yet to be integrated. To ensure the quality of this narrative review, we adhered to the SANRA scale [88] and tried to provide unbiased and reliable information.

## Figures and Tables

**Figure 1 healthcare-13-00357-f001:**
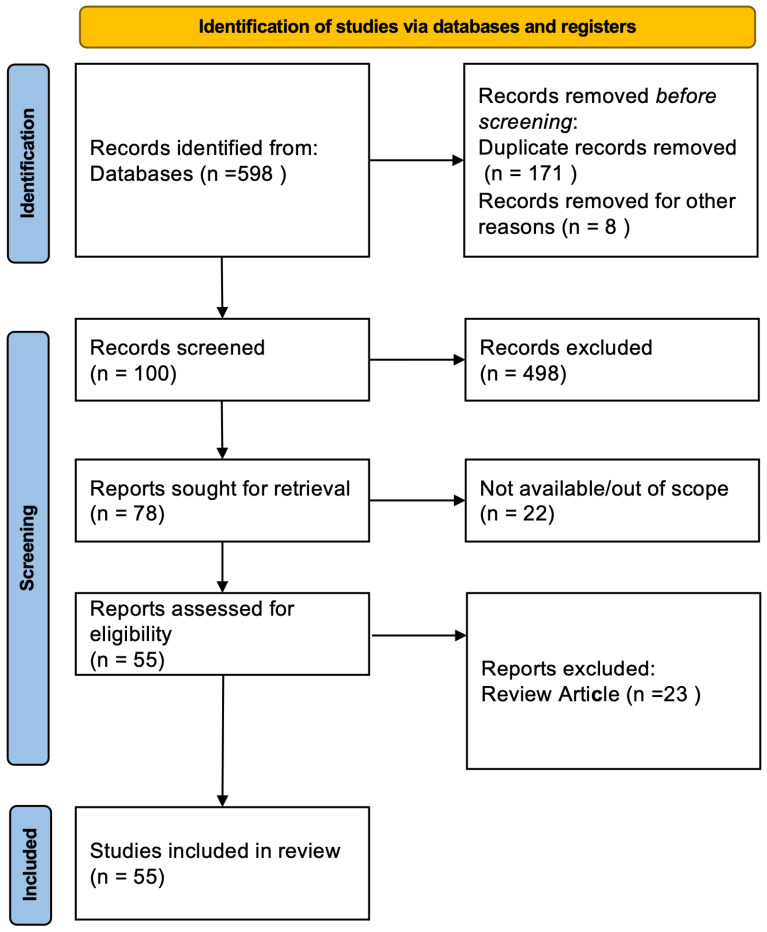
Flow diagram of literature selection process.

**Table 1 healthcare-13-00357-t001:** Literature search strategy for Medline via PubMed and CINAHL.

	Search Query	Results	Remarks
#1	“Nursing Homes”[MeSH Terms] OR “long term care”[MeSH Terms] OR “Homes for the Aged”[MeSH Terms] OR “nursing home*”[Title/Abstract] OR “long term care”[Title/Abstract] OR “Residential Aged Care Facility”[Title/Abstract] OR “old age home*”[Title/Abstract] OR “Senior Housing”[Title/Abstract]	99,596	nursing homes/long-term care
#2	“Accidental Falls”[MeSH Terms] OR “fall*”[Title/Abstract]	268,521	fall
#3	“Forecasting”[MeSH Terms] OR “Accident Prevention”[MeSH Terms] OR “Risk Assessment”[MeSH Terms] OR “Geriatric Assessment”[MeSH Terms] OR “Digital Technology”[MeSH Terms] OR “Big Data”[MeSH Terms] OR “Data Mining”[MeSH Terms] OR “Datasets as Topic”[MeSH Terms] OR “Data Science”[MeSH Terms] OR “Data Collection”[MeSH Terms] OR “prediction*”[Title/Abstract] OR “prevention*”[Title/Abstract] OR “assessment*”[Title/Abstract] OR “digital technology*”[Title/Abstract] OR “Digital Electronics”[Title/Abstract] OR “Big Data”[Title/Abstract] OR “Data Mining”[Title/Abstract] OR “Text Mining”[Title/Abstract] OR “dataset*”[Title/Abstract] OR “data set*”[Title/Abstract] OR “data science*”[Title/Abstract] OR “data analytic*”[Title/Abstract] OR “Data Collection”[Title/Abstract]	5,585,191	Assessment/Prevention
#4	“Aged”[MeSH Terms] OR “Aged”[Title/Abstract]	4,143,060	Aged
#5	#1 AND #2 AND #3 AND #4	1881	
#6	#5 AND (2019:2024[pdat])	470	2019–2024
Search Strategy: PubMed	((“Nursing Homes”[MeSH Terms] OR “long term care”[MeSH Terms] OR “Homes for the Aged”[MeSH Terms] OR “nursing home*”[Title/Abstract] OR “long term care”[Title/Abstract] OR “Residential Aged Care Facility”[Title/Abstract] OR “old age home*”[Title/Abstract] OR “Senior Housing”[Title/Abstract]) AND (“Accidental Falls”[MeSH Terms] OR “fall*”[Title/Abstract]) AND (“Forecasting”[MeSH Terms] OR “Accident Prevention”[MeSH Terms] OR “Risk Assessment”[MeSH Terms] OR “Geriatric Assessment”[MeSH Terms] OR “Digital Technology”[MeSH Terms] OR “Big Data”[MeSH Terms] OR “Data Mining”[MeSH Terms] OR “Datasets as Topic”[MeSH Terms] OR “Data Science”[MeSH Terms] OR “Data Collection”[MeSH Terms] OR “prediction*”[Title/Abstract] OR “prevention*”[Title/Abstract] OR “assessment*”[Title/Abstract] OR “digital technolog*”[Title/Abstract] OR “Digital Electronics”[Title/Abstract] OR “Big Data”[Title/Abstract] OR “Data Mining”[Title/Abstract] OR “Text Mining”[Title/Abstract] OR “dataset*”[Title/Abstract] OR “data set*”[Title/Abstract] OR “data science*”[Title/Abstract] OR “data analytic*”[Title/Abstract] OR “Data Collection”[Title/Abstract]) AND (“Aged”[MeSH Terms] OR “Aged”[Title/Abstract])) AND (2019:2024[pdat])		
Search Strategy: CINAHL	((MH “Nursing Homes+”) OR (MH “Long Term Care”) OR TI (“Nursing Home*” OR “Long Term Care” OR “Residential Aged Care Facility” OR “Old Age Home*” OR “Senior Housing”) OR AB (“Nursing Home*” OR “Long Term Care” OR “Residential Aged Care Facility” OR “Old Age Home*” OR “Senior Housing”)) AND ((MH “Accidental Falls”) OR TI fall* OR AB fall*) AND ((MH “Forecasting”) OR (MH “Fall Prevention (Iowa NIC)”) OR (MH “Risk Assessment”) OR (MH “Geriatric Assessment+”) OR (MH “Digital Technology+”) OR (MH “Data Mining+”) OR (MH “Data Science”) OR (MH “Data Collection+”) OR TI (Prediction* OR Prevention* OR Assessment* OR “Digital Technolog*” OR “Digital Electronics” OR “Big Data” OR “Data Mining” OR “Text Mining” OR Dataset* OR “Data set*” OR “Data Science*” OR “Data Analytic*” OR “Data Collection”) OR AB (Prediction* OR Prevention* OR Assessment* OR “Digital Technolog*” OR “Digital Electronics” OR “Big Data” OR “Data Mining” OR “Text Mining” OR Dataset* OR “Data set*” OR “Data Science*” OR “Data Analytic*” OR “Data Collection”)) AND ((MH “Aged+”) OR TI aged OR AB aged)		

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
