# Peer review of "Fall Risk Assessment and Prevention Strategies in Nursing Homes: A Narrative Review"

_healthcare, 2025, doi:10.3390/healthcare13040357_

Round 1
Reviewer 1 Report
Comments and Suggestions for Authors
General Comments:
The manuscript “Predicting Falls in Nursing Homes: A Review” presents an analysis of data from articles related to fall risk assessment tools and fall prevention programs. This narrative review is not poorly written (general impression), especially given the context of the lack of a clear framework for such articles. There are just some suggestions for evaluating narrative reviews - SANRA. The authors have taken this into account, which is worth noting.
On the other hand, there are some questions/doubts:
i) This is a narrative review - so in its nature it is subjective and consequently low on the hierarchy of scientific evidence.
ii) What new contributions does this review make? And how does this article extend the knowledge presented in systematic reviews available in databases?
A PubMed search using the same search string (as the authors) with the additional condition “review[title]” found at least 4 articles on a similar topic. There was also a 2022 article on guidelines for preventing falls in nursing homes:
1: Shao L, Shi Y, Xie XY, Wang Z, Wang ZA, Zhang JE. Incidence and Risk Factors of Falls Among Older People in Nursing Homes: Systematic Review and Meta-Analysis. J Am Med Dir Assoc. 2023 Nov;24(11):1708-1717. doi: 10.1016/j.jamda.2023.06.002.
2: Gulka HJ, Patel V, Arora T, McArthur C, Iaboni A. Efficacy and Generalizability of Falls Prevention Interventions in Nursing Homes: A Systematic Review and Meta-analysis. J Am Med Dir Assoc. 2020 Aug;21(8):1024-1035.e4. doi: 10.1016/j.jamda.2019.11.012.
3: Dawson R, Suen J, Sherrington C, Kwok W, Pinheiro MB, Haynes A, McLennan C,
Sutcliffe K, Kneale D, Dyer S. Effective fall prevention exercise in residential aged care: an intervention component analysis from an updated systematic review. Br J Sports Med. 2024 May 31;58(12):641-648. doi: 10.1136/bjsports-2023-107505.
4: Huang X, Jiang Y, Liu Y, Shen L, Pan J, Zhang Y. Influencing factors of falls among older adults in Chinese retirement institutions: A systematic review and meta-analysis. PLoS One. 2023 Dec 27;18(12):e0296348. doi: 10.1371/journal.pone.0296348.
5: Schoberer D, Breimaier HE, Zuschnegg J, Findling T, Schaffer S, Archan T. Fall prevention in hospitals and nursing homes: Clinical practice guideline. Worldviews Evid Based Nurs. 2022 Apr;19(2):86-93. doi: 10.1111/wvn.12571.
Some of these papers are referenced by the Authors…
iii) The use of the term prediction/prediction throughout the concept of the article is somewhat misleading. In the reviewer's opinion, the authors were rather analyzing tools related to fall risk assessment (in fact, this is in line with the objectives of L61-L62). Prediction is an appropriate term when statistical models are considered, but not in a practical situation - you can't predict that a person will fall today, for example, but you can say that a person has a very high risk of falling.
iv) The review seems to be somewhat flat and superficial “there are some risk assessment tools and there are some prevention programs.” - and all that.... no other information is analyzed.
Detailed remarks:
1. Title.
i) it does not reflect what was analyzed in the review, ii) as written above, “prediction” is not the appropriate term here, iii) it should be demonstrated expresis verbis that the article is a narrative review.
2. Abstract.
i) please make an order with statements “Abstract: Background/Objectives:” or “Results: Conclusions:”, ii) there are no results reported, iii) aims are different than in main text.
3. Introduction
i) it would be nice to strengthen arguments for providing this review, ii) L42-44 – it’s surprising to see additional aim in the middle of the Introduction (by the way, it’s again a little bit next to aims showed at the end of this section),
4. Methods
i) reviewer’s impression is that some information is still repeated as the 5 year period, nursing home settings or aim, ii) L71-73 – aim again! And again a little bit different that previous notation, iii) L78 – PubMed is a platform not a database – it should be rather “Medline via PubMed”, iv) what was the search strategy for CINAHL? it’s doubtful that the Authors used the same search string as CINAHL doesn’t operate MeSH terms, v) it would be nice to see what kind of studies (study design) were included.
5. Results and discussion
No specific comments on these sections. The authors have done what they planned, presented a synthesis of the data and their interpretation. The reviewer does not intend to interfere with these elements. However, as indicated above, the overall analysis seems somewhat superficial.
Reviewer 2 Report
Comments and Suggestions for Authors
The article provides a comprehensive review of fall prediction and prevention methods in nursing homes, focusing on studies conducted between 2019 and 2024. It emphasizes the importance of addressing the unique challenges in long-term care settings, such as limited resources, high dependency levels, and the need for continuous care. While the article is well-structured and highlights critical factors such as risk assessment tools, digital monitoring technologies, and multifactorial prevention strategies, there are some suggestions for the authors.
1. The abstract lacks a complete summary of the main results. The authors should add its.
2. The article briefly mentions that studies focusing on nursing home settings between 2019 and 2024 were included. However, the specific inclusion and exclusion criteria are not fully explained. The article does not explicitly state the types of studies reviewed (e.g., randomized controlled trials, cohort studies, systematic reviews). The authors have to provide a detailed description of inclusion and exclusion criteria.
3. There is no mention of a systematic process for assessing the quality or risk of bias of the studies included in the review. This omission limits the ability to evaluate the reliability of the findings presented. The authors should incorporate a standardized quality assessment tool to evaluate the methodological rigor of the studies.
4. The conclusion is somewhat generic. The authors should highlight key actionable insights or a clear call to action for researchers and practitioners; they might include a summary of how the findings can directly improve nursing home care practices.
Round 2
Reviewer 1 Report
Comments and Suggestions for Authors
I appreciate that the authors addressed all my comments. Congratulations.